# Position: From Crowdsourcing to Crowd-LLM-Sourcing and LLM-Sourcing

**Jiyi Li** [1]

## Abstract

Crowdsourcing has been widely adopted for large-scale data collection and problem solving, yet its outcomes are often noisy and inconsistent, making quality control and aggregation central concerns. Meanwhile, Large Language Models (LLMs) have shown strong capabilities in generation, annotation, evaluation, and reasoning. These developments can be framed as an emerging paradigm at the intersection of crowdsourcing and LLMs, encompassing two directions: (1) Crowd-LLM-Sourcing, where humans and LLMs jointly participate in workflows, and (2) LLM-Sourcing Inspired by Crowdsourcing, where crowdsourcing principles guide LLM-driven generation, annotation, evaluation, and inference. Many existing studies on LLMs overlook decades of prior work in crowdsourcing, even though the two domains are grounded in closely related principles on some topics. The central position of this paper is that, in scenarios where an LLM can be regarded as an LLM worker, LLM research should draw upon the rich body of crowdsourcing literature. At the same time, LLM workers differ from human workers in their correlated errors and context-dependent capabilities. This paper therefore highlights not only the relevance of crowdsourcing, but also the need to adapt its mechanisms for collective intelligence with model-based agents.

## 1. Introduction

Crowdsourcing has been widely adopted as an effective and scalable method for collecting large volumes of human-generated answers in both data acquisition and problem-solving tasks (e.g., (Snow et al., 2008; Bernstein et al., 2015)). By distributing work across a large population, crowdsourcing systems are able to harness the collective intelligence of individuals with diverse backgrounds, experiences, and skill sets. This diversity is a core strength: it enables the exploration of multiple perspectives, supports creative problem solving, and allows complex tasks to be decomposed into manageable units that can be processed in parallel.

Despite these advantages, crowdsourcing outcomes often suffer from inherent challenges, including data inconsistency, annotation noise, and substantial variation in worker reliability, expertise, and motivation (e.g., (Dawid & Skene, 1979; Whitehill et al., 2009)). Differences in workers' understanding, effort, and intent can lead to conflicting or low-quality responses. As a result, quality assurance mechanisms, such as worker modeling, task design, redundancy, aggregation, and incentive structures, have become central concerns in both crowdsourcing research and real-world deployments. The success of a crowdsourcing system depends not only on scale, but also on how effectively it manages uncertainty and variability in human performance. Importantly, these mechanisms are not limited to simple redundancy or majority voting; they form a statistical and procedural toolkit for modeling worker ability, item difficulty, bias, dependence, task assignment, aggregation, and multi-stage quality control.

Meanwhile, Large Language Models (LLMs) have recently demonstrated strong capabilities in generation, annotation, evaluation, and reasoning. Empirical studies show that LLMs can match or even outperform average crowd workers on a range of annotation tasks (e.g., (Gilardi et al., 2023; Cegin et al., 2023)), and a growing body of work explores LLMs as annotators, evaluators, or judges (e.g., (He et al., 2024a; Gu et al., 2025)). These developments can be framed as an emerging paradigm at the intersection of crowdsourcing and LLMs. Two key and complementary directions can be identified within this emerging paradigm.

**Crowd-LLM-Sourcing:** Human crowds and LLMs collaborate within shared crowdsourcing workflows. In this setting, LLMs may assist workers by providing suggestions, drafts, or feedback, while humans verify, refine, and correct model outputs. Conversely, human judgments can guide, supervise, or calibrate LLM behavior. Such hybrid systems aim to combine the creativity, contextual understanding, and

---

[1]Graduate School of Information Science and Technology, Hokkaido University, Sapporo, Japan. Correspondence to: Jiyi Li <garfieldpigljy@gmail.com>.

ethical awareness of humans with the speed, consistency, and scalability of LLMs.

**LLM-Sourcing Inspired by Crowdsourcing:** Principles and methodologies developed in crowdsourcing are applied to optimize LLM-driven data generation, annotation, evaluation, and inference. Here, LLMs are treated as a population of "workers" with varying behaviors across prompts, temperatures, model variants, or checkpoints. Concepts such as *redundancy, worker modeling, task assignment, aggregation, and quality control* can be reinterpreted to manage and improve LLM outputs. This view differs from treating LLM agents merely as components in an engineering pipeline. Instead, it adopts a population-level perspective: multiple humans, models, prompts, samples, or model variants are regarded as imperfect agents whose outputs should be assigned, calibrated, compared, aggregated, and verified under explicit assumptions about ability, difficulty, diversity, and dependence.

**However, many existing studies on LLMs overlook decades of prior work in crowdsourcing, even though the two domains are grounded in closely related principles on some topics.** Both involve uncertain agents producing imperfect outputs, both require mechanisms for quality control and aggregation, and both benefit from modeling agent behavior, diversity, bias, and reliability. Ignoring this literature risks reinventing established ideas, such as redundancy, staged verification, worker/item modeling, and aggregation beyond majority voting, while missing opportunities to adapt these ideas systematically to model-based agents.

This paper provides a concise review of traditional crowdsourcing, systematically compares crowdsourcing and LLM-based systems, and introduces how the field is shifting from the traditional crowdsourcing paradigm toward broader and more general paradigms of Crowd-LLM-Sourcing and LLM-Sourcing. **The central position is that, in scenarios where an LLM can be regarded as an *LLM worker*, LLM research should draw upon and be inspired by the extensive crowdsourcing literature. LLM studies can benefit from established approaches to modeling worker characteristics, behavior, reliability, bias, and uncertainty, as well as from proven strategies for task design, incentive alignment, redundancy, and result aggregation.**

Crucially, this paper does not claim that LLMs are simply interchangeable with human workers. LLM workers differ from humans in fundamental ways: their errors are often correlated rather than independent, their behavior is highly prompt-dependent, and their "abilities" are not stable traits but context-conditioned capabilities. As a result, several assumptions underlying classical crowdsourcing break in multi-LLM settings. A central goal of this paper is therefore not only to align crowdsourcing mechanisms with LLM research, but to expose where these analogies fail and to articulate how they should be adapted, for example through correlation-aware aggregation, capability-conditioned routing, and so on.

## 2. Crowdsourcing

As discussed in the introduction, the central challenge of crowdsourcing lies in managing noisy and heterogeneous workers. To address this challenge, the field has developed a rich set of principles and mechanisms, most notably redundancy, worker modeling, aggregation, and multi-stage workflows, that transform unreliable individual responses into reliable collective outcomes. The goal in this section is not to provide an exhaustive survey of crowdsourcing, but to extract a set of design principles that are directly relevant to Crowd-LLM-Sourcing and LLM-Sourcing.

This section reviews a representative subset of foundational crowdsourcing studies that embody these principles. Although the literature is vast, the works selected here are sufficient to convey the core ideas that are most relevant to, and inspiring for, LLM research in the following sections.

### 2.1. Quality Control for Categorical Labels

This subsection first introduces annotations for categorization tasks and the corresponding label aggregation methods for categorical labels. This is one of the most typical annotation tasks that can be used to assign categories when collecting ground truth for classification, or to select answers from multiple choices when constructing knowledge bases. For categorical annotation, each worker assigns a category label to an object, and the goal is to infer the latent true labels from noisy worker responses. Table 1.(a) gives an example. Crowdsourcing methods often also estimate worker characteristics, such as abilities or confusion patterns.

For the categorical labels, *majority voting* (Snow et al., 2008) is a simple and effective answer aggregation method but only assigns equal weights to all workers. Because it only assigns equal weights to all crowd workers and labels, the quality of the aggregated answers is not stable because of diverse quality. Researchers also proposed more sophisticated models. Some approaches jointly estimate worker ability and true answers using the expectation-maximization (EM) algorithm (Dawid & Skene, 1979), the maximum entropy principle (Zhou et al., 2012), and Bayesian inference (Liu et al., 2012; Venanzi et al., 2014). Some models incorporate task difficulty (Whitehill et al., 2009), and their Bayesian treatments (Wauthier & Jordan, 2011; Bachrach et al., 2012). There are also other recent works, and we just list some of them (Li et al., 2017; 2018b; Li & Kashima, 2017; Li, 2019; Kawase et al., 2019; Li et al., 2020; Li & de Rijke, 2023; Li et al., 2023; Yang et al., 2024; Li, 2024b;

| (a) **Categorization**
Estimated Category | $y_1^2 = c_3$: worker $a^2$ judges $o_1$ as category $c_3$; $y_2^3 = c_2$; $y_3^1 = c_1$; $y_3^2 = c_2$; ...
Categorical label aggregation: $y_1 = c_2$; $y_2 = c_2$; $y_3 = c_2$; $y_4 = c_3$; $y_5 = c_3$ |
| --- | --- |
| (b) **Pairwise Preference Comparison**
Estimated Ranking | $o_1 \succ_{a^1} o_2$: worker $a^1$ prefers $o_1$ to $o_2$; $o_2 \succ_{a^2} o_3$; $o_3 \succ_{a^2} o_4$; $o_1 \succ_{a^1} o_4$; ...
Pairwise preference aggregation result: $o_1 \succ o_2 \succ o_3 \succ o_4 \succ o_5$ |
| (c) **Pairwise Similarity Comparison**
Estimated Clusters | $o_1 \sim_{a^1} o_2$: worker $a^1$ judges $o_1$ and $o_2$ *similar*; $o_3 \neq_{a^1} o_4$; $o_1 \sim_{a^2} o_3$; $o_2 \neq_{a^2} o_5$; ...
Pairwise similarity label aggregation result: $o_1 \sim o_2 \sim o_3$, $o_4 \sim o_5$ |

*Table 1.* Examples of categorization, pairwise preference and pairwise similarity tasks.

2025; Zhang et al., 2024b;a). Besides answer aggregation approaches, there is another type of approach that directly trains the classification models with the noisy crowd labels (Rodrigues & Pereira, 2018; Li et al., 2022; Lu et al., 2023).

A representative label aggregation method for categorical labels is GLAD (Whitehill et al., 2009). The original GLAD method utilizes an Expectation Maximization approach (EM) to obtain maximum likelihood estimates of the potential parameters, i.e., estimated labels $\hat{\mathcal{Y}}$, worker abilities $\Theta$, and task easiness $\Gamma = \{\gamma_i\}_{i=1}^n$. It can be reformulated into one objective function. The probability $q_i^l$ that a worker $a^l$ assigns the correct label to an object $o_i$ can be formulated as $q_i^l = 1/\left(1 + (\mathcal{K} - 1)e^{-\theta^l \gamma_i}\right)$. Assuming that a worker assigns an incorrect answer uniformly at random. The crowd labels $y_i^l$ can be formulated as a $\mathcal{K}$-dimensional one-hot binary vector, only one of its elements is 1, $\mathbf{y}_i^l = (0, \ldots, 0, 1, 0, \ldots, 0)^\top \in \{0,1\}^{\mathcal{K}}$. The estimated labels $\hat{y}_i$ can be formulated as a $\mathcal{K}$-dimensional one-hot binary vector $\hat{\mathbf{y}}_i = (0, \ldots, 0, 1, 0, \ldots, 0)^\top \in \{0,1\}^{\mathcal{K}}$. $\mathbf{y}_i^{l\top} \hat{\mathbf{y}}_i$ can compute the correctness of label $\mathbf{y}_i^l$ by worker $a^l$ to object $o_i$. The objective function based on log-likelihood of the crowd labels can be reformulated as follows. $\mathcal{L}_{glad} = \sum_{i,l} \left( \mathbf{y}_i^{l\top} \hat{\mathbf{y}}_i \log q_i^l + (1 - \mathbf{y}_i^{l\top} \hat{\mathbf{y}}_i) \log \left( \frac{1 - q_i^l}{\mathcal{K} - 1} \right) \right)$.

Moreover, although Item Response Theory (IRT) (Hambleton et al., 1991) originated in educational measurement rather than in crowdsourcing, it is grounded in the same core principle as GLAD: jointly modeling worker (or examinee) ability and item difficulty to explain observed responses.

The classical models above are often presented under conditional-independence assumptions: given the latent true label, worker ability, and item difficulty, worker responses are treated as independent observations. This assumption is useful, but it is not universal in crowdsourcing research. Later work has explicitly modeled structured dependence among workers. For example, community-based Bayesian aggregation models (Venanzi et al., 2014) assume that workers can belong to latent communities with similar confusion patterns, while correlation-aware aggregation models directly exploit dependencies among workers (Li et al., 2019). This distinction is important for our later discussion of LLM-Sourcing: if LLMs from similar model families, training pipelines, or prompting strategies behave

like correlated worker communities, GLAD-style objectives should be adapted rather than transferred verbatim.

## 2.2. Quality Control for Other Types of Annotations

Beyond categorical labels, crowdsourcing literature has developed a wide range of aggregation and modeling methods for diverse forms of human judgments. In this subsection, we briefly review diverse types of labels.

### 2.2.1. PAIRWISE PREFERENCE COMPARISON

In practice, humans often find it easier to provide relative judgments between instances than to assign absolute labels or scores. For example, it is usually more intuitive to compare two images and decide which one is more appealing than to rate each image on a fixed numerical scale. This form of feedback, known as pairwise preference comparison, reduces cognitive load and is widely used in tasks involving subjective evaluation. The goal of the preference task is to obtain a ranking list reflecting the relative preference among the objects. Table 1.(b) provides an example with pairwise preference comparison labels and aggregated results. Because the number of all candidate object pairs is huge for pairwise preference comparisons, only a small subset of labels is required; each object pair is annotated by multiple workers; each worker only annotates a small subset. The target is to obtain accurate ranking results using a limited number of pairwise comparison labels.

For pairwise preference comparison labels (Cattelan, 2012), a typical solution is the Bradley-Terry model (Bradley & Terry, 1952) and its various extensions or generalizations to diverse settings (Causeur & Husson, 2005; Davidson, 1970; Hunter, 2004; Chen & Joachims, 2016b;a; Chen et al., 2013; Raman & Joachims, 2014; Li et al., 2018a; Baba et al., 2020; Jin et al., 2020; Zuo et al., 2020; Li, 2022; Li et al., 2021; Zhang et al., 2022). There are also other types of methods, e.g., matrix completion (Yi et al., 2013; Oh et al., 2015).

One typical model for the ranking task based on pairwise preference comparisons is CROWDBT (Chen et al., 2013), which extends Bradley-Terry (BT) model (Bradley & Terry, 1952) to the crowdsourced setting by incorporating worker reliability into preference estimation.

### 2.2.2. OTHER TYPES OF HUMAN JUDGMENTS

For representation learning, it is often easier to compare two images and decide whether they are similar than to assign them fine-grained categories that require professional knowledge. This type of feedback is referred to as pairwise similarity comparison. The goal of the similarity task is to obtain proximity relations among the objects. The binary questions for the workers can be "whether these two objects are similar or not?". Table 1.(c) provides an example with pairwise similarity comparison labels and aggregated results. For the pairwise similarity comparison task, the object embeddings are learned from high-dimensional feature representations or pairwise similarity comparison labels by preserving the neighborhoods or pairwise similarities (Gomes et al., 2011; Yi et al., 2012; Nguyen et al., 2023; Ariu et al., 2024).

In addition, there are also many other types of annotations. These include triplet similarity labels (van Der Maaten & Weinberger, 2012; Lu et al., 2023), which are widely used in representation learning and metric learning; numerical annotations (Li et al., 2014), where workers provide real-valued or ordinal estimates; and more complex structured data, such as bounding boxes and taxonomy paths (Meir et al., 2024), which arise in tasks involving spatial localization or hierarchical reasoning. There is also a growing body of work on aggregating free-form text annotations (Li, 2020; 2024a), which addresses challenges such as semantic variability, paraphrasing, and subjective interpretation.

Furthermore, there are also a few existing works concentrating on multiple tasks (Zhou et al., 2019; Li, 2025) for crowdsourcing. Zhou et al. (2019) merged multiple datasets on the homogeneous tasks and proposed an optimization framework for dual learning from task and worker. Li (2025) proposed general label aggregation approach for composite crowd tasks by bridging homogeneous or heterogeneous single-crowd-tasks in composite-crowd-tasks with worker ability constraint satisfaction and relaxed optimization.

These studies demonstrate that crowdsourcing is not limited to simple classification tasks, but supports a rich spectrum of human judgments with varying structures, semantics, and uncertainty characteristics. Each data type introduces distinct challenges for quality control and aggregation, motivating specialized probabilistic models, optimization methods, and learning-based approaches. Together, they provide a comprehensive toolkit for reasoning about noisy, heterogeneous, and subjective human inputs. LLM systems rarely produce only categorical labels: they generate rankings, pairwise preferences, similarity judgments, rationales, structured outputs, and free-form text. Therefore, the relevant crowdsourcing toolkit should not be restricted to categorical labels, but should also include aggregation methods for preferences, structures, and open-ended judgments.

### 2.3. Crowdsourcing Pipelines

In crowdsourcing area, single-stage frameworks typically collect all annotations from crowd workers in a single round. While simple and efficient, such designs provide limited flexibility for requesters to incorporate diverse mechanisms, such as verification, refinement, or targeted rework, to improve data quality. To address these limitations, a variety of multi-stage crowdsourcing frameworks have been proposed, in which complex tasks are decomposed into structured pipelines composed of multiple coordinated stages.

For example, Partition-Map-Reduce (Kittur et al., 2011) is designed for essay writing: the task is first partitioned into subtopics, then multiple workers independently draft content for each part, and finally the partial outputs are merged and refined into a coherent essay. Creation-Review (Baba & Kashima, 2013) adopts a two-stage pipeline for creative tasks such as image description and logo design, where one group of workers generates initial artifacts and another group reviews, critiques, and improves them. More importantly, this line of work shows that multi-stage pipelines can model the quality of different roles separately: the ability of creators and the ability of reviewers are distinct sources of uncertainty that should both be estimated. Find-Fix-Verify (Bernstein et al., 2015) targets text editing tasks such as shortening long documents: workers first identify problematic segments (Find), then propose revisions (Fix), and finally validate the corrections (Verify). Pair-Compare-Sort (Cheng et al., 2015) addresses sorting problems by decomposing them into a sequence of pairwise comparisons, enabling reliable global ordering through local judgments.

Together, these lines of research demonstrate that multi-stage pipelines provide a powerful abstraction for managing complexity and quality in human computation, offering rich design principles that can directly inform the construction of LLM-based and hybrid Crowd-LLM workflows.

## 3. Crowd-LLM-Sourcing

LLMs have recently demonstrated strong capabilities in generation, annotation, evaluation, and reasoning. This section focuses on Crowd-LLM-Sourcing, where LLMs and human crowds jointly participate in crowdsourcing workflows.

Recently, the capability of LLMs on data annotation tasks has attracted interest from researchers. Because LLMs are cheaper than crowd workers for annotating the instances (e.g., (Gilardi et al., 2023) reported that ChatGPT is about twenty times cheaper than MTurk in their experiments), a natural question is whether LLMs can replace, assist, or complement different components of crowdsourcing workflows. Veselovsky et al. (Veselovsky et al., 2023) found that the crowd workers on MTurk have been recently using LLMs to complete the crowdsourcing tasks. Some works

verified this issue with the average performance of individual crowd workers and LLMs on some specific tasks by collecting new datasets for their target tasks (Zhu et al., 2023; Gilardi et al., 2023; Törnberg, 2023; Cegin et al., 2023; He et al., 2024a). For instance, Gilardi et al. (Gilardi et al., 2023) demonstrated that ChatGPT outperforms crowd workers in terms of both accuracy and consistency across several text classification tasks, including stance detection, topic categorization, and framing identification. Similarly, Törnberg et al. (Törnberg, 2023) reported that ChatGPT-4 surpassed both crowd workers and domain experts in classifying political tweets.

These LLM studies mainly concentrate on the average performance of individual crowd workers. However, this comparison only captures one component of a crowdsourcing system. In practice, the eventually collected annotations are usually not crowd answers themselves, but aggregated answers produced from multiple noisy and diverse answers. Therefore, the scenarios involving crowd answer aggregation need further study. Several recent works (Li, 2024b; He et al., 2024b) studied the scenario of answer aggregation on the crowd categorical labels and closed-ended text answers (Li, 2024a). The investigation in Li (2024b) based on crowdsourcing datasets shows that (1) LLM workers outperform the average quality of responses from typical crowd workers; (2) low-ability crowd workers provide very low quality labels; (3) high-ability crowd workers outperform LLM workers; (4) aggregating both crowd labels and LLM labels can achieve higher-quality results than using only crowd labels or only LLM labels; (5) crowd labels have higher diversity than LLM labels, which may be one potential reason of the high quality of the hybrid Crowd-LLM aggregated labels. This finding suggests that the value of human crowds is not limited to producing labels with higher average accuracy. Human workers can also provide non-model diversity: they bring different experiences, interpretations, real-world priors, and error patterns that are not easily reproduced by sampling from similar LLMs. This diversity is especially important when LLM outputs are correlated, overconfident, or shaped by shared training data and architectural biases.

In addition, Baumann et al. (2025) found that intentional LLM hacking is strikingly simple. By replicating 37 data annotation tasks from 21 published social science studies, the results show that, with just a handful of prompt paraphrases, virtually anything can be presented as statistically significant. Conversely, studies have indicated that when data collection relies exclusively on LLM-generated outputs and models are repeatedly trained on this synthetic data, the result can be model collapse. This issue arises from the reduction of data diversity and the reinforcement of pre-existing biases (Shumailov et al., 2024; Christoforou et al., 2025). Incorporating human crowd input provides a safeguard against such degradation.

In summary, although LLMs offer advantages in lowering costs and ensuring consistency, they cannot yet fully replace expert-level crowd contributors. Instead, a promising direction lies in integrating LLMs and human workers within crowdsourcing workflows. Future Crowd-LLM-Sourcing systems should explicitly study when to rely on LLMs, when to involve human workers, and how to aggregate human, LLM-generated, and LLM-assisted labels under different reliability and diversity assumptions. The first part of Table 2 aligns crowdsourcing mechanisms with LLM research paradigms from the viewpoint of Crowd-LLM-Sourcing.

## 4. LLM-Sourcing Inspired by Crowdsourcing

Beyond hybrid workflows, the same crowdsourcing principles can also be applied to purely LLM-driven systems. Crowdsourcing techniques such as *redundancy, aggregation, task design, and quality control* can be adapted to LLM-driven data generation, annotation, evaluation, and inference. Multiple LLMs or multiple outputs from the same LLM can be treated analogously to "workers" with methods such as consensus, weighting, routing, or adjudication ensuring robust results. Some LLM-based studies have proposed ideas similar or related to those in the crowdsourcing literature, e.g., *LLM ensemble (Chen et al., 2025), multi-LLM annotation (Qi et al., 2026), multi-LLM routing (Hu et al., 2024; Feng et al., 2025; Song et al., 2025), training (Zhao et al., 2025), reasoning (Wang et al., 2023; Motwani et al., 2025), LLM-as-a-judge (Lu et al., 2024; Sun et al., 2025; Gu et al., 2025), chain-of-thought prompting (Wei et al., 2022), problem-solving pipelines (Wu et al., 2025)*. Some cite crowdsourcing studies or related studies, while some overlook it. For example, some "new" LLM reasoning paradigms are rediscovering classical crowdsourcing mechanism, including redundancy, worker modeling, task routing, staged pipelines, and aggregation, often without explicitly grounding in the crowdsourcing literature.

Two representative examples are discussed below, but the adaptation of crowdsourcing techniques for multi-LLM systems is not limited to them. The second and third parts of Table 2 provide a systematic alignment between core mechanisms in crowdsourcing and emerging paradigms in LLM research. The goal is not to claim that all multi-LLM or agentic architectures originate from crowdsourcing. Rather, crowdsourcing is used as a principled lens for organizing design problems that many LLM systems already face: how to allocate tasks to imperfect agents, how to estimate agent capability and item difficulty, how to aggregate conflicting outputs, how to detect low-quality responses, and how to design multi-stage workflows. This perspective helps move from ad-hoc engineering heuristics toward explicit assumptions and reusable modeling principles.

This manuscript uses the terms crowdsourcing and LLM-

| Crowdsourcing Studies | LLM Studies | Alignment |
|---|---|---|
| Hybrid human+AI workflows | Crowd-LLM-Sourcing (LLMs as "LLM workers") | LLMs can outperform average workers but not necessarily top workers; hybrid designs combine scalability, reliability, diversity (Li, 2024a;b). |
| Multiple workers / redundancy | Mixture of multiple LLMs, multi-sample decoding (e.g., self-consistency) | Treat multiple LLMs or multiple samples as a "crowd" of workers; next generation active learning: mixture of LLMs-in-the-Loop (Qi et al., 2026); exploit diversity via redundancy and consensus (Wang et al., 2023). |
| Worker ability + item difficulty modeling (e.g., GLAD, IRT ) | Multi-LLM routing; IRT-style routing; model selection; calibration; capability modeling | Route queries to the most suitable "LLM worker"; analogous to assigning tasks to workers by skill (Hu et al., 2024; Feng et al., 2025). IRT-style modeling resembles classic worker/item models for disentangling ability vs difficulty (Song et al., 2025). |
| Aggregation of noisy labels | Majority vote; verifier models; judge-based selection; | Classical aggregation maps to majority vote, learned aggregators, and verifier; judge paradigms (Chen et al., 2025; Gu et al., 2025). |
| "Minority-but-correct" phenomena in aggregation | Training aggregation beyond majority vote | "Majority is not always right" is well-known in crowdsourcing; LLM work revisits it for learning better aggregators (Zhao et al., 2025). |
| Quality control (gold tests, review, adjudication) | LLM-as-a-judge; adjudication; consensus + reranking | Crowdsourcing quality control mechanisms translate into LLM judging, adjudication, and weighting strategies (Gu et al., 2025; Lu et al., 2024; Sun et al., 2025). |
| Multi-stage workflow; task decomposition pipelines | Chain-of-thought prompting; agentic pipelines | Multi-stage crowdsourcing pipelines align with LLM "chains" and multi-LLM reasoning pipelines (Wu et al., 2025). |
| Human pipeline (e.g., Find-Fix-Verify) | Agent training and reasoning pipeline (e.g., Generator-Verifier-Refiner) | Direct structural analogy: identify issues → propose fixes → verify/refine (Motwani et al., 2025). |

*Table 2.* Aligning crowdsourcing mechanisms with LLM research paradigms under the broader "Crowd-LLM-Sourcing" and "LLM-Sourcing" view.

Sourcing. A closely related family of viewpoints is often framed in terms of human-in-the-loop and LLM-in-the-loop systems. These perspectives emphasize the role of humans or models as supervisory or auxiliary components within a learning or decision-making pipeline, rather than as "workers" in a collective. While the terminology differs, the underlying concern is similar: how to combine human and machine intelligence in a principled and effective manner. This formulation highlights the structural parallels between human crowds and collections of LLMs, making explicit that both can be viewed as populations of imperfect agents whose outputs should be coordinated, filtered, and aggregated. This population view is what distinguishes LLM-Sourcing from a generic "loop" or "agent pipeline" perspective: the key object of study is not only the behavior of one model, but the distribution, dependence, disagreement, and aggregation of many imperfect model outputs.

### 4.1. Adapting Quality Control Approaches for Multi-LLM Systems

**Multiple workers / redundancy.** Crowdsourcing has long relied on redundancy, i.e., assigning the same task to multiple workers, to counter noise and variability. LLM re-

search mirrors this idea through mixtures of multiple models and multi-sample decoding, such as self-consistency (Wang et al., 2023). Here, multiple outputs from one or more LLMs form a "crowd" whose diversity can be exploited to improve robustness. Recent work further extends this idea into next-generation active learning, where a mixture of LLMs operates in the loop (Qi et al., 2026), echoing the role of diverse human workers in adaptive data collection. However, redundancy in LLM-Sourcing should not be equated with simple repetition. Because similar models or prompts may produce correlated outputs, this analogy requires the diversity-aware and correlation-aware adaptations discussed in Section 4.4.

**Worker ability and item difficulty modeling.** In crowdsourcing, tasks are assigned based on estimated worker skill, cost, and suitability, often using models that disentangle worker ability from item difficulty (e.g., GLAD (Whitehill et al., 2009), IRT (Hambleton et al., 1991)). Multi-LLM systems increasingly mirror this paradigm by treating models as heterogeneous "LLM workers" and routing each query to the most suitable one. From this perspective, model selection becomes a model (worker) - query (item) modeling or task-assignment problem: queries are routed according

to predicted model competence (Hu et al., 2024; Feng et al., 2025). IRT-style routing (Song et al., 2025) makes this analogy explicit by modeling both query difficulty and model capability, directly paralleling classical worker-item frameworks. This brings established crowd modeling principles, e.g., capability estimation, calibration, and skill-aware assignment, into the design of multi-LLM pipelines. The crowdsourcing perspective therefore reframes routing not as a heuristic model-selection rule, but as a worker-item modeling problem: a routing system should estimate both the capability of each LLM worker and the difficulty, ambiguity, or domain requirements of each query. This also suggests that capability should be modeled as context-conditioned rather than as a single global score for each model.

**Aggregation of noisy labels.** Aggregating imperfect human labels is central to crowdsourcing, with majority vote and probabilistic models as standard tools. LLM systems face an analogous challenge when combining multiple generated answers (Chen et al., 2025; Gu et al., 2025). Majority voting, verifier models, and judge-based selection directly correspond to crowd aggregation strategies, transforming noisy LLM outputs into a single reliable decision. More importantly, the analogy should extend beyond majority voting. For LLM-Sourcing, aggregation must account for dependence among LLM workers; Section 4.4 discusses how community-based and correlation-aware crowd models suggest a path for such adaptation.

**Minority-but-correct phenomena.** Crowdsourcing has long recognized that the majority is not always right; minority answers can be correct when workers have unequal expertise. Recent LLM work revisits this phenomenon by training aggregation mechanisms beyond simple voting (Zhao et al., 2025). This rediscovery highlights how LLM research is encountering the same fundamental issues of heterogeneity and bias that motivated advanced crowd aggregation decades ago.

**Quality control.** Crowdsourcing employs gold questions, review, and adjudication to ensure quality. LLM systems now adopt analogous mechanisms through LLM-as-a-judge frameworks, consensus checking, and reranking (Gu et al., 2025; Lu et al., 2024; Sun et al., 2025). These methods reinterpret human quality control pipelines for model outputs, reinforcing the view of LLMs as workers subject to verification and oversight.

### 4.2. Adapting Human-in-the-loop Problem Solving Approaches for LLM Pipelines

**Multi-stage workflows.** Complex crowd tasks are often decomposed into multi-stage pipelines, where subtasks are solved sequentially. LLM research reflects this structure through Chain-of-thought prompting (Wei et al., 2022) or agentic reasoning pipelines (Wu et al., 2025). Both

paradigms recognize that structured decomposition enables more reliable problem solving than monolithic execution.

**Crowd pipelines and LLM pipelines.** Frameworks such as Find-Fix-Verify (Bernstein et al., 2015) exemplify how human workflows separate error detection, correction, and validation. LLM research mirrors this with, e.g., Generator-Verifier-Refiner pipelines (Motwani et al., 2025), for reasoning and training. The structural correspondence is direct: identify issues, propose solutions, and verify or refine them. The deeper lesson is not merely that both systems use a verifier. Crowdsourcing pipelines suggest that different stages should be modeled as distinct roles with different failure modes: a generator may be fluent but inaccurate, a verifier may be conservative or biased, and a refiner may introduce new errors while fixing old ones. For LLM pipelines, this implies that the reliability of generators, verifiers, and refiners should be estimated separately, rather than assuming that adding a verification stage automatically improves quality. This alignment underscores how multi-agent LLM reasoning recapitulates classical human-in-the-loop problem-solving designs.

### 4.3. Human Workers vs. LLM workers

The mappings above reveal strong structural parallels. However, these parallels are only approximate. Although both humans and LLMs can be regarded as "workers" in a collective system, they differ fundamentally in their characteristics and behavioral dynamics. Human workers are heterogeneous in expertise, motivation, attention, and bias. Their behavior is shaped by incentives, fatigue, learning effects, and social context. Errors made by humans are often idiosyncratic: different workers fail in different ways, and variability is a defining feature of the crowd.

In LLM-Sourcing, the term "LLM worker" can refer to several related but distinct units: a model instance selected from different model families, a sampled output from the same model under different decoding settings, or a role-specific agent in a pipeline such as a generator, verifier, or refiner. These worker types require different crowdsourcing strategies. Model-family workers call for capability estimation and routing; sampled-output workers call for diversity-aware aggregation; role-specific agents call for stage-wise quality control and role-specific reliability estimation.

LLM workers, by contrast, exhibit a distinct profile. Given the same prompt and configuration, an LLM behaves deterministically, while diversity arises primarily from sampling strategies, prompt variations, model versions, or architectural differences. LLMs do not tire, lose motivation, or strategically respond to incentives. Instead, they display systematic properties such as prompt sensitivity, calibration errors, confident hallucinations, and crucially correlated failure modes across similar models or configurations. Errors

are often not random but stem from shared training data, inductive biases, and architectural constraints. These differences imply that LLM workers cannot be treated as a drop-in replacement for human workers. While both can be modeled as imperfect agents in a collective system, the nature of their imperfections is qualitatively different. Human crowds are noisy but diverse; LLM crowds are consistent but prone to shared blind spots. As a result, crowdsourcing techniques cannot be transferred to LLM-based systems verbatim. They should be reinterpreted and adapted to account for the distinctive properties of model-based agents. Thus, while human crowds often provide naturally occurring diversity, LLM crowds require deliberately constructed diversity across prompts, models, training lineages, decoding strategies, and human oversight.

### 4.4. When Crowdsourcing Analogies Break: What Fails and How to Adapt

The analogies between crowdsourcing and LLM-Sourcing illuminate powerful structural similarities, but they also expose where classical assumptions break down. Crowdsourcing methods are built upon a set of implicit premises about human workers that do not hold for LLMs. Understanding these failure points is essential for developing principled LLM-Sourcing systems.

Redundancy and independence. Traditional crowdsourcing relies on the assumption that worker errors are largely independent. Redundancy is effective because different humans make different mistakes; aggregation reduces variance and suppresses noise. LLMs violate this assumption. Models trained on similar data and architectures tend to fail in the same regions of the input space, reproduce the same hallucinations, and reinforce shared biases. Naïve redundancy, such as sampling the same model multiple times with minor prompt variations, can therefore amplify systematic errors rather than cancel them. Effective "LLM redundancy" should be engineered to promote structural diversity, across model families, training regimes, prompting strategies, and decoding policies, rather than relying on repetition alone. The adaptation is to move from quantity-based redundancy to diversity-aware redundancy and correlation-aware aggregation.

Worker modeling and stability. Crowdsourcing models such as GLAD assume that each worker has relatively stable latent traits, e.g., ability, bias, or reliability, that generalize across tasks. LLM behavior, however, is highly context-dependent. Performance varies not only by task but by prompt formulation, instruction style, and interaction history. An LLM worker does not possess a single, fixed "ability" parameter; its competence is a function over a high-dimensional prompt and task space. Classical worker models should therefore be reformulated as capability surfaces conditioned on both items and contexts, and routing decisions should jointly consider instance properties and prompt configurations.

Quality control and verification. In human crowdsourcing, gold questions, review, and adjudication are effective because workers adapt to oversight and incentives. LLMs are non-strategic: errors reflect model limitations rather than effort. Consequently, simply replacing human reviewers with LLM-as-a-judge can create circular failure modes, where generation and evaluation share the same blind spots. Using heterogeneous models can reduce this risk, but it does not guarantee independence. Robust quality control requires asymmetric designs, e.g., separating generators and judges across heterogeneous models or across human-LLM boundaries, to avoid self-reinforcement.

Aggregation and minority signals. Crowdsourcing theory recognizes that minorities can be correct when expertise is unevenly distributed. In LLM systems, minority outputs may arise from stochastic decoding or prompt instability rather than superior competence. Distinguishing "minority-but-correct" from "minority-but-random" becomes a central challenge. Aggregation should therefore go beyond frequency, incorporating signals such as model provenance, calibration, cross-model disagreement structure, and confidence estimates. Community-based and correlation-aware aggregation methods suggest that model families, training lineages, or prompting regimes can be treated as correlated worker groups rather than independent voters.

These failures do not diminish the relevance of crowdsourcing research; they sharpen it. Crowdsourcing provides the conceptual vocabulary, e.g., redundancy, worker modeling, aggregation, staged workflows, but LLM systems require these concepts to be re-instantiated under new assumptions: correlated failures instead of independent noise, context-dependent capability instead of stable ability, and systematic bias instead of random error. The core research challenge is not to imitate crowdsourcing, but to generalize it.

## 5. Call to Action

The convergence between crowdsourcing and LLM research suggests a research agenda for principled Crowd-LLM and LLM-Sourcing systems. Rather than treating LLMs as isolated solvers, these systems view humans, models, samples, and agents as imperfect contributors whose outputs must be selected, coordinated, verified, and aggregated. Crowdsourcing is not the only theoretical foundation for such systems, but it provides useful principles for making these assumptions explicit.

**For LLM researchers.** Treat multi-model, multi-sample, and multi-agent systems as algorithmic crowds whose outputs may be noisy, biased, and correlated. Crowdsourc-

ing research offers useful tools for this setting, including worker/item modeling, redundancy design, task assignment, aggregation beyond majority voting, and quality control. The key challenge is to adapt these tools to LLM-specific properties, such as context-dependent capability, prompt sensitivity, and shared failure modes, rather than assuming that existing crowd models can be transferred directly.

**For crowdsourcing researchers.** Study LLMs as worker-like but non-human agents. Classical concepts such as ability, reliability, bias, task difficulty, and worker dependence remain useful, but their meanings change in model-based systems. For example, an LLM's ability may depend on prompt, context, decoding strategy, model family, and task domain, while its errors may be correlated with those of other models trained on similar data. This creates opportunities to extend crowdsourcing theory toward prompt-conditioned ability modeling, correlation-aware aggregation, and role-specific quality estimation in LLM pipelines.

**For data researchers.** Move beyond the dichotomy of "synthetic versus human" data. Instead of choosing between LLMs and people, design data pipelines that orchestrate both. LLMs can generate, pre-filter, draft, and annotate at scale, while humans can verify, refine, contribute diverse judgments, and provide real-world reference points. From this perspective, data creation is not a binary choice between human and synthetic sources, but a coordinated process for balancing scalability, diversity, reliability, and oversight.

**For dataset curators and benchmark designers.** Shift evaluation from single-model performance to system-level behavior. Develop benchmarks that measure how well ensembles, routed models, and hybrid human-LLM systems coordinate, recover from errors, handle correlated failures, and aggregate diverse outputs. What matters is not only how strong a model is in isolation, but how effective a system is under collective operation.

**For system builders and practitioners.** Design workflows in which generation, verification, refinement, and aggregation are treated as distinct roles with measurable failure modes. Adding more agents or samples should not be assumed to improve reliability; systems should explicitly consider diversity, dependence, task difficulty, and the independence of verification signals. In hybrid systems, humans and LLMs should be orchestrated as complementary sources of scale, judgment, and error correction.

## 6. Alternative Views

An alternative viewpoint argues that LLMs are fundamentally different from human crowds and therefore require entirely new methods. LLMs and humans clearly differ in important ways, including their error patterns, diversity, adaptivity, incentives, and modes of failure. LLMs may exhibit highly correlated mistakes, prompt sensitivity, and confident hallucinations, while humans show variability driven by expertise, attention, and motivation. However, these differences do not invalidate the relevance of crowdsourcing research. Instead, they motivate adapting crowdsourcing ideas rather than discarding them. Concepts such as redundancy, worker modeling, task routing, staged workflows, and aggregation remain valuable abstractions; what changes is how these concepts are instantiated for model-based agents. Indeed, systematically characterizing how LLM workers differ from human workers in their characteristics and behaviors, and how classical methods should be modified accordingly, constitutes a rich and underexplored research direction in its own right.

Another alternative view is that these issues are already covered by existing multi-agent, scalable oversight, or verifier-based LLM research, and that increasingly capable LLMs may eventually reduce the need for human crowds. These perspectives are complementary rather than contradictory. Crowdsourcing is not the sole foundation for multi-agent LLM systems, and human crowds are not always necessary. Rather, crowdsourcing offers a population-level methodology for reasoning about imperfect agents, dependence, aggregation, quality control, and workflow design. At the same time, the risks of pure LLM-Sourcing, including correlated failures, shared blind spots, prompt sensitivity, and reduced diversity, suggest that hybrid human-LLM workflows remain an important part of the design space.

## 7. Conclusion

This paper reviewed crowdsourcing mechanisms that are particularly relevant to Crowd-LLM-Sourcing and LLM-Sourcing. The central argument is that crowdsourcing offers more than simple redundancy, majority voting, or verifier-based selection: it provides a broader toolkit for modeling agent ability, item difficulty, dependence, aggregation, task assignment, and staged quality control. At the same time, LLM workers differ from human workers in their correlated errors, prompt sensitivity, capability instability, and shared training-induced biases. The resulting research agenda is not to transfer crowdsourcing mechanisms verbatim, but to adapt them for collective intelligence with model-based agents. This perspective aims to clarify when human, model, and hybrid collectives should be used, rather than treating any one of them as universally preferable.

## Acknowledgements

This work was supported by JSPS KAKENHI Grant Number JP23K28092. The author thanks Hisashi Kashima, Yukino Baba, Xiaotian Lu, Jingzheng Li, Hailong Sun, and many other collaborators in prior crowdsourcing work.

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
