# OpenReview forum: "Position: From Crowdsourcing to Crowd-LLM-Sourcing and LLM-Sourcing"
_ICML.cc/2026/Position_Paper_Track — ICML 2026 Position Paper Track regular_

### Official Review · Reviewer_1Jkg · 2026-03-12

**Significance:** 3
**Argument Clarity:** 2
**Rating:** 4
**Confidence:** 4

**Questions:**

Section 4.4 points out that "LLM-as-a-judge may share the blind spots of the generator". If heterogeneous models are adopted to break this cycle (e.g., GPT-4 as the generator and Claude as the judge), how to explain that systematic preference consistency still exists among different LLM judges? Does this imply that blind spots are universal across models?

**Alternative Views Section:**

Yes

**Compliance With Llm Reviewing Policy A Conservative:**

Affirmed.

**Discussion Potential:**

3

**Final Justification:**

The overall quality of the work meets the ICML bar, so I am maintaining my positive score.

**Paper Summary:**

This paper proposes a new paradigm of Crowd-LLM-Sourcing and advocates for the transfer of the theoretical framework accumulated over decades in crowdsourcing to large language model (LLM) research. It systematically reviews quality control mechanisms, multi-stage workflows, and aggregation methods in crowdsourcing, and establishes correspondences between these elements and LLM techniques such as Self-Consistency, LLM-as-a-Judge, and Chain-of-Thought. The core argument is that LLMs can be regarded as LLM workers, yet traditional crowdsourcing mechanisms need to be re-adapted to account for LLMs' unique characteristics in terms of error correlation, prompt sensitivity, and capability instability.

**Position:**

Yes

**Position In Title:**

Yes

**Related Work:**

2

**Strengths And Weaknesses:**

Strengths:

- The paper accurately identifies the phenomenon of "reinventing the wheel" in current LLM research. Many seemingly novel LLM ensemble methods are in fact variants of classic crowdsourcing techniques, yet they lack citations to relevant literature. This observation carries important warning value for the research community.

- Rather than simply applying crowdsourcing theories, the authors explicitly point out the fundamental differences between LLM workers and human workers (correlated errors vs. independent errors, prompt dependence vs. capability stability), and propose corresponding adaptation directions based on these differences (Section 4.3-4.4).

Weaknesses:

- The paper refers to LLMs as three distinct types of "workers" in different contexts. However, the paper fails to discuss the corresponding differences in crowdsourcing strategies for each type.

- Crowdsourcing itself faces bottlenecks such as high costs, high latency, and difficulties in scaling to professional domains. Yet the paper does not fully explore whether LLM-Sourcing introduces new risks while addressing these bottlenecks,

**Support:**

3

---

> ### Author Rebuttal · Authors · 2026-03-31
>
> We sincerely thank reviewer 1Jkg for the thoughtful and constructive feedback and for recognizing that our work addresses the **"reinventing the wheel"** phenomenon in current LLM research. Your insightful queries regarding worker types, new risks, and the universality of blind spots strike at the core of our position. We find this feedback **perfectly aligned with our argument**, and we will incorporate these discussions into our revised manuscript to strengthen the proposed research agenda.
> 1. **Systematic Preference Consistency and Universal Blind Spots.** The reviewer raises a profound point regarding whether these blind spots are "universal" across model families. On the one hand, our paper identifies the phenomenon of "shared blind spots" and "correlated failures" based on existing evidence. As shown in (Li, ICASSP 2024b), crowd labels possess significantly higher diversity and are less correlated than LLM labels. On the other hand, we agree that the extent of systematic preference consistency across highly heterogeneous models (like GPT-4 vs. Claude) remains an open and critically important research question. LLMs often share training data and architectural constraints, their errors may be correlated rather than independent.
> We think this is **an important direction for future research**. Our main point is that diversity should not be defined only by model brand. Using GPT-4 and Claude together does not automatically give us meaningful diversity. We need to examine what different model families can and cannot do, and identify the areas where they fail in similar ways. This is also why **Crowd-LLM Collaboration** matters. Human judgment brings perspectives and real-world reference points that models often do not have.
>
> 2. **On the "Three Distinct Types of Workers" and Their Adaptations.** While our manuscript does not explicitly label these "three types", we interpret your observation, based on the reviewer’s summary of "LLM techniques such as Self-Consistency, LLM-as-a-Judge, and Chain-of-Thought", as pointing to three different functional roles that LLMs can play. Below, we briefly explain how crowdsourcing strategies should be adapted for each role.
> - **Individual annotators or judges.**
> When an LLM is used as a single judge, the main concern is how to evaluate and control its decisions. In this setting, the focus should be on **adjudication and quality control**, rather than on aligning incentives as we do with human workers. We also argue for asymmetric designs, so that the same model is not simply asked to check or reinforce its own reasoning, which could create circular failure modes.
> - **Ensembles or model populations.**
> When multiple LLM outputs are treated as a kind of “crowd,” as in self-consistency, the key issue becomes **diversity**. Unlike human crowds, model diversity does not arise naturally. It should be deliberately designed, e.g., by combining different model families or decoding strategies to reduce correlated errors.
> - **Sequential agents in pipelines.**
> When LLMs are used as consecutive steps in a workflow, such as in agentic pipelines, the key issue is **task decomposition**. Following the Find-Fix-Verify principle, we argue that error detection and error correction should be separated, so that a model is less likely to repeat or reinforce its own mistakes across steps.
>
> 3. **New Risks Introduced by LLM-Sourcing.** The reviewer correctly notes that while LLM-Sourcing addresses human bottlenecks like cost and latency, it is not a "free lunch" and introduces distinct new risks.
> - **Correlated failures.**
> Unlike human crowds, LLM workers often fail in similar ways because they share training data and architectural constraints. As a result, simply adding more LLMs may amplify systematic errors rather than reduce them.
> - **Diversity loss and model collapse.**
> As discussed in Section 3, relying only on LLM-Sourcing risks model collapse. Without the diversity contributed by humans, systems may reinforce shared biases and lose the benefits of crowd wisdom.
> - **Prompt sensitivity and capability instability.**
> LLM performance is highly sensitive to prompts and context, so its capability is not a stable trait. This makes traditional worker-modeling methods less effective without the adaptations.
>
> We will further clarify and expand our discussion on these LLM-Sourcing risks in the revised manuscript to ensure that the trade-offs between efficiency (addressing crowdsourcing bottlenecks) and reliability (managing LLM-Sourcing risks) are explicitly discussed.

---

> > ### Author Rebuttal · Reviewer_1Jkg · 2026-04-02
> >
> > My concerns have been adequately addressed.

---

### Official Review · Reviewer_rysB · 2026-03-13

**Significance:** 3
**Argument Clarity:** 4
**Rating:** 5
**Confidence:** 4

**Questions:**

1. Are there deeper connections of LLMs and crowdsourcing? Are there ideas from crowdsourcing that haven't been applied to LLMs yet?
2. Are there ways to make models less uniform and correlated if their training data is similar?

**Alternative Views Section:**

Yes

**Compliance With Llm Reviewing Policy A Conservative:**

Affirmed.

**Discussion Potential:**

4

**Final Justification:**

I increase my score as the author's rebuttal provides additional insight and really reinforces the value of revisiting prior crowd-sourcing research in light of models uniquely trained on the internet and then with verifier-driven RL.

**Paper Summary:**

This paper offers the perspective of crowdsourcing design on LLM design. They highlight both how ideas from crowdsourcing do and don't apply to LLMs in particular the main challenge in LLMs have correlate failures making averages not helpful. Second they look at LLM aided crowdsourcing. This approach looks a lot like scalable oversight and offer ways to augment and combine cheap LLM assistance with cheap human labeling.

**Position:**

Yes

**Position In Title:**

Yes

**Related Work:**

3

**Strengths And Weaknesses:**

Strength:

1. The authors acknowledge from the get-go the difference in diversity and failure correlation of humans vs LLMs.
2. Provide interesting insight on how combining LLMs and humans can be better than either individually.
3. The analogy of crowdsourcing is thought provoking and interesting juxaposition.

Weakness:
1. The examples of using crowdsourcing for LLMs seems straight forward and in retrospect not insightful. Other fields of research also suggest the power of verifiers and majority voting. Without a deeper connection, it may be worth focusing this position paper on just aspect (2) using LLMs + humans to design unique reliable systems.

**Support:**

4

---

> ### Author Rebuttal · Authors · 2026-03-31
>
> We sincerely thank reviewer rysB for the thoughtful and constructive feedback. The reviewer’s suggestions regarding deeper connections and methods to reduce correlation are excellent extensions of our work. These challenges are central to the **"New Research Agenda"** we propose, and we will incorporate these important points into our revised manuscript.
> 1. **Untapped Ideas from Crowdsourcing Literature.** The reviewer asks for deeper connections and untapped ideas. We argue that the crowdsourcing literature offers a **sophisticated toolkit** that goes far **beyond basic heuristics like verifiers and majority voting.**
> - **Statistical Ability Estimation in Pipelines.** The reviewer notes that other fields of research also suggest **the power of verifiers**. In fact, even in multi-stage pipelines that seem "straightforward," crowdsourcing research provides deep insights. For example, in the Creation-Review pipeline proposed by **(Baba et al., KDD 2013)**, the focus is not on simple verification. Instead, it involves **statistical quality estimation of the abilities of both the "generator (Creation)" and the "verifier (Review)"**.
> - **Beyond Majority Voting.** The reviewer notes that other fields of research also suggest **the power of majority voting**. In crowdsourcing, **majority voting is merely a foundational baseline**. The core contribution of the field lies in sophisticated probabilistic models like **D&S (Dawid & Skene, Applied Statistics 1979), and GLAD (Whitehill et al., NIPS 2009)**. Unlike simple voting, these models **jointly model worker ability (reliability/bias) and item difficulty** to explain observed responses.
> - While simple classification is common, crowdsourcing also offers a **comprehensive toolkit** for much more complex human judgments. For example, techniques for aggregating **complex structures such as bounding boxes, taxonomy paths (Meir et al., AAAI 2024), or free-form text (Li, SIGIR 2020)**, which are standard in crowdsourcing, could significantly enhance multi-LLM reasoning for spatial or hierarchical tasks.
> - Another example is the **crowdsourcing studies related to the worker correlation**, which will be introduced in Point 2.
> 2. **Addressing Model Uniformity and Correlation.** The concern about model uniformity is central to our argument that **"naïve redundancy" fails** for LLMs. Because LLMs often share training data and architectural constraints, their errors are correlated rather than independent. To handle this issue, we consider several candidate solutions.
> - The crowdsourcing researchers have developed sophisticated models to account for structured dependencies among workers.  For example, as cited in our paper, **(Venanzi et al., WWW 2014)** is a **community-based Bayesian label aggregation model**, which assumes that crowd workers conform to a few different types, where **each type represents a group of workers with similar confusion matrices**. This serves as a direct roadmap for adaptation: e.g., **"model families"** (e.g., Llama-based models) are treated as **"worker communities"**. Identifying such existing toolkits for adaptation is a core contribution of our paper. Another example is “Exploiting Worker Correlation for Label Aggregation in Crowdsourcing” (Li et al., ICML 2019) which proposes **an enhanced Bayesian classifier combination model for exploiting worker correlation.** Besides, there are also other studies, and **LLM research can be inspired by the crowdsourcing literature for this correlation issue of LLMs.** We will improve the presentation on this issue.
> - **Crowd-LLM Collaboration.** From **our perspective as researchers in the crowdsourcing field**, this systematic consistency among models further validates the necessity of the **Crowd-LLM Collaboration** paradigm we propose. While the community may prefer pure LLM-Sourcing for scalability, we think that human judgment serves as the essential **resource to provide the diversity**. As shown in **(Li, ICASSP 2024b)**, crowd labels possess **significantly higher diversity** and are less correlated than LLM labels. This inherent human diversity is one of the reasons why hybrid Crowd-LLM systems can achieve higher quality than pure model-based systems. It can also provide a vital safeguard against **model collapse**.
> - **Diversity of Regimes.** Possibly, effective redundancy should move beyond simple repetition and toward diversity across **model families, training regimes, and decoding policies.**
> 3. **A Pluralistic Theoretical View.** We wish to clarify that we do not suggest that all multi-agent architectures must exclusively originate from crowdsourcing. We view crowdsourcing as **one of several robust theoretical frameworks** that can provide structured insights. While our paper focuses on this perspective due to its direct structural parallels with LLM "workers," we recognize and welcome other complementary theories as part of the broader solution space.

---

> > ### Author Rebuttal · Reviewer_rysB · 2026-04-04
> >
> > I really appreciate the additional references the authors provided and recommend these be added to the paper. I also really like the insights around model uniformity and think this should be a bigger part of the discussion. Especially as agentic systems become more popular more clarity on how to take advantage of different model providers is going to be a timely research direction.

---

### Official Review · Reviewer_ftUN · 2026-03-13

**Significance:** 2
**Argument Clarity:** 3
**Rating:** 4
**Confidence:** 3

**Questions:**

1. The paper mentions classical crowdsourcing relies on the assumption of independent worker errors, while LLMs errors are somehow correlated. Is there a specific solution towards how an aggregation algorithm (e.g., like the GLAD mentioned in the paper) should be modified?
2. How does "Crowd-LLM-Sourcing" differ from the existing advanced multi-agent routing and collaboration architectures?

**Alternative Views Section:**

Yes

**Compliance With Llm Reviewing Policy A Conservative:**

Affirmed.

**Discussion Potential:**

3

**Final Justification:**

The authors have done well in addressing my concerns and also those of other reviewers. I have increased my score.

**Paper Summary:**

This paper proposes “Crowd-LLM-Sourcing”, suggesting in scenarios where an LLM can be regarded as an LLM worker, LLM research should draw upon the rich body of crowdsourcing literature. Two main directions are categorized in this paper: human-LLM hybrid workflows and multi-LLM systems inspired by crowdsourcing principles (e.g., routing, aggregation, and redundancy). Moreover, the paper emphasizes that while LLMs can act as "workers," they are different from human workers, speficially due to LLMs’ deterministic nature, context-dependent capabilities, and highly similar failure modes due to shared training data and architectures.

**Position:**

Yes

**Position In Title:**

No

**Related Work:**

2

**Strengths And Weaknesses:**

Strengths:

1.	The paper draws a clear structural comparations between classical crowdsourcing models (like GLAD and Bradley-Terry) and existing paradigms such as LLM-as-a-judge.

2.	Although the paper suggest treating LLM as a ‘worker’, the authors correctly identify the flow under this assumption: errors from human are generated independently, while biases and errors from LLMs reasoning are similar due to their shared architecture or training data.

Weaknesses:

1.	There is a lack of mathematical proof: The paper points out that classical crowdsourcing objective (like Equation 2) fail due to the correlated / similar errors of LLMs, but there is not fix raised to the model .

2.	The call-to-action part is somehow overstated. It’s correct, but when building multiagent architecture, basically current developers already know that static or unified models often fail and that conditioned routing is required when agents are interacting with each other, so it’s not a new opintion.

3.	Although the paper claims the cases when crowdsourcing analogies cannot be applied on LLM systems, there is no specific examples proofing or demonstrate the statements.

**Support:**

2

---

> ### Author Rebuttal · Authors · 2026-03-31
>
> We sincerely thank reviewer ftUN for the thoughtful and constructive feedback. We find the reviewer’s observations regarding the complexity of error correlations and the need for specific adaptations to be **perfectly aligned with our central position**. These points highlight essential **future research directions** within the **"New Research Agenda"** we advocate. We will incorporate these insights into our revised manuscript to provide a more robust theoretical roadmap.
>
> 1. **On the Role of a Position Paper vs. Mathematical Proof.** The reviewer notes a lack of a specific "mathematical fix" for the proposed paradigm. We would like to clarify that as a **Position Paper**, our primary objective is to **define the problem space and establish a principled framework** for a previously fragmented field. We believe that providing a systematic conceptual alignment is the foundation upon which **future mathematical proofs and optimized algorithms will be built**.
>
> 2. **Handling Correlated Errors with Advanced Crowdsourcing Models.** The reviewer correctly points out that while classical crowdsourcing relies on independent worker errors, LLM errors are correlated. For the question of “Is there a specific solution towards how an aggregation algorithm (e.g., like the GLAD mentioned in the paper) should be modified”, in fact, **not all crowdsourcing research assumes independence**. On the one hand, classical crowdsourcing methods such as D&S and GLAD rely on independent worker errors. On the other hand, the crowdsourcing researchers have already developed sophisticated models to account for structured dependencies among workers.
> For example, as cited in our paper, (Venanzi et al., WWW 2014) is a **community-based Bayesian label aggregation model**, which assumes that crowd workers conform to a few different types, where **each type represents a group of workers with similar confusion matrices**. This serves as a direct roadmap for adaptation: e.g., **"model families"** (e.g., Llama-based models) are treated as **"worker communities"**. Identifying such existing toolkits for adaptation is a core contribution of our paper. Another example is “Exploiting Worker Correlation for Label Aggregation in Crowdsourcing” (Li et al., ICML 2019) which proposes **an enhanced Bayesian classifier combination model for exploiting worker correlation.** Besides, there are also other studies, and **LLM research can be inspired by the crowdsourcing literature for this correlation issue of LLMs.** We will improve the presentation on this issue.
>
> 3. **Distinction from existing advanced multi-agent routing and collaboration architectures.**
> - **Success of re-instantiation. IRT-Router (Song et al., ACL 2025)** is a perfect example of our position in action, it explicitly uses **Item Response Theory** from crowdsourcing to model model-capability and item-difficulty, proving that aligning these fields leads to more interpretable routing.
> - **Structural analogy.** We show that advanced reasoning chains (e.g., **Generator-Verifier-Refiner (Motwani et al., COLM 2025)** are **direct re-instantiations** of classical crowdsourcing workflows like **Find-Fix-Verify (Bernstein et al., Commun. ACM 2015) **. This allows designers to apply proven crowdsourcing principles of task decomposition and quality control to LLM systems.
> - **Population view.** Unlike "loop" views that treat agents as auxiliary components, we view them as **a population of imperfect agents** whose outputs should be coordinated through statistical aggregation and quality control.
>
> 4. **Addressing the "Call to Action" and Novelty.** The reviewer points out that current developers already recognize the failure of static models and the need for conditioned routing. We agree. On the other hand, our position is that while **the need is recognized**, the **principled methodology** is often missing or reinvented without grounding in established literature.
> - **From Tricks to Paradigm.** Current multi-agent architectures often rely on "engineering tricks" and empirical trial-and-error. Our framework advocates for a shift toward theoretically grounded Crowd-LLM systems.
> - **Systematic Methodology.** While a developer might intuitively implement a router, our work points to rigorous, decades-old crowdsourcing methodologies, such as **Item Response Theory** or **Bayesian community modeling**, to disentangle agent ability from task difficulty in a mathematically consistent way.
> - **A Pluralistic Theoretical View.** Furthermore, we wish to clarify that we do not suggest that all multi-agent routing or collaboration architectures must exclusively originate from crowdsourcing. We view crowdsourcing as **one of several robust theoretical frameworks** that can provide structured insights. While our paper focuses on the crowdsourcing perspective due to its direct structural parallels with LLM "workers," we recognize and welcome other complementary theories as part of the broader solution space.

---

> > ### Author Rebuttal · Reviewer_ftUN · 2026-04-04
> >
> > The rebuttal is comprehensive and have addressed my questions and concerns. As such I will increase my score.

---

### Decision · Program_Chairs · 2026-04-30

**Decision:**

Accept (regular)

**Comment:**

The paper makes a timely and thought-provoking conceptual contribution by connecting two literatures that are often treated separately, and the reviews suggest that both the core position and the rebuttal were persuasive, with reviewers’ concerns substantially resolved and all three reviewers ending at positive or borderline-positive recommendations. I recommend acceptance of the paper.